# Using natural language processing to identify acute care patients who lack advance directives, decisional capacity, and surrogate decision makers

Jiyoun Song[1], Maxim Topaz[1,2,3], Aviv Y. Landau[2,4], Robert Klitzman[5,6], Jingjing Shang[1], Patricia Stone[1], Margaret McDonald[3], Bevin Cohen[7,8]*

1 Columbia University School of Nursing, New York, New York, United States of America, 2 Data Science Institute, Columbia University, New York, New York, United States of America, 3 Visiting Nurse Service of New York, New York, New York, United States of America, 4 Columbia School of Social Work, New York, New York, United States of America, 5 Vagelos College of Physicians and Surgeons, Columbia University, New York, New York, United States of America, 6 Mailman School of Public Health, Columbia University, New York, New York, United States of America, 7 Center for Nursing Research and Innovation, Mount Sinai Health System, New York, New York, United States of America, 8 Department of Geriatric and Palliative Medicine, Icahn School of Medicine at Mount Sinai, New York, New York, United States of America

* bevin.cohen@mountsinai.org

**Data Availability Statement:** The data underlying the results presented in the study are available from the Medical Information Mart for Intensive

## Abstract

The prevalence of patients who are Incapacitated with No Evident Advance Directives or Surrogates (INEADS) remains unknown because such data are not routinely captured in structured electronic health records. This study sought to develop and validate a natural language processing (NLP) algorithm to identify information related to being INEADS from clinical notes. We used a publicly available dataset of critical care patients from 2001 through 2012 at a United States academic medical center, which contained 418,393 relevant clinical notes for 23,904 adult admissions. We developed 17 subcategories indicating reduced or elevated potential for being INEADS, and created a vocabulary of terms and expressions within each. We used an NLP application to create a language model and expand these vocabularies. The NLP algorithm was validated against gold standard manual review of 300 notes and showed good performance overall (F-score = 0.83). More than 80% of admissions had notes containing information in at least one subcategory. Thirty percent (n = 7,134) contained at least one of five social subcategories indicating elevated potential for being INEADS, and <1% (n = 81) contained at least four, which we classified as high likelihood of being INEADS. Among these, n = 8 admissions had no subcategory indicating reduced likelihood of being INEADS, and appeared to meet the definition of INEADS following manual review. Among the remaining n = 73 who had at least one subcategory indicating reduced likelihood of being INEADS, manual review of a 10% sample showed that most did not appear to be INEADS. Compared with the full cohort, the high likelihood group was significantly more likely to die during hospitalization and within four years, to have Medicaid, to have an emergency admission, and to be male. This investigation demonstrates potential for NLP to identify INEADS patients, and may inform interventions to enhance advance care planning for patients who lack social support.

Care III (MIMIC-III) database at (https://mimic.mit.edu/). These are third party data not owned or collected by the authors. The authors did not have any special access privileges that others would not have. All instructions for use of the data are provided on the MIMIC-III website.

**Funding:** Our study was funded by National Institute of Nursing Research [NINR] (R21 NR019319), "Improving ethical care for patients who are incapacitated with no evident advance directives or surrogates (INEADS)". The funders had no role in study design, data collection and analysis, decision to publish, or preparation of the manuscript.

**Competing interests:** The authors have declared that no competing interests exist.

## Introduction

Honoring patient preferences is a tenet of medical ethics and a measure of healthcare quality [1]. Still, the majority of Americans do not have advance directives or healthcare proxies to guide their care when they lack the capacity to make decisions for themselves [2, 3]. Nearly 65% of adults in the United States have no advance directives documenting their wishes or appointing surrogate decision makers [2]. At the same time, 33–65% of hospitalized adults lack decisional capacity [4–6]. This leaves patients vulnerable to receiving care that is unaligned with their values, goals, and preferences, and places care teams in the difficult position of having to search for patient advocates and navigate an ethically fraught plan of care [7–9]. In the absence of advance directives and appointed healthcare proxies, care teams often identify family or friends who can provide information about patients' wishes and values [10]. However, sometimes patients become decisionally incapacitated without any advance directives or personal contacts known to the healthcare team [11]. We refer to such patients as being Incapacitated with No Evident Advance Directives or Surrogates (INEADS).

The number of patients meeting these criteria is expected to rise in the coming decades due to an aging population, a growing number of patients with dementia, and a rise in the number of seniors who live alone [12, 13]. However, estimating the number of patients who are INEADS has been challenging due to a near total lack of data describing this phenomenon. A single study conducted in 2003–2004 found that 16% of ICU patients were INEADS, though these findings reflect the experience of a single unit in an urban public hospital over a short period before Medical/Physician/Provider Orders for Life Sustaining Treatment (MOLST/POLST) programs were widely adopted in the United States [14, 15]. The scarcity of information around this topic may be due in part to the fact that these characteristics are not routinely nor accurately captured in structured healthcare data, meaning that efforts to collect such information have historically required resource intensive methods.

The availability of modern data science methods like natural language processing (NLP) provide a novel opportunity to understand the characteristics and prevalence of INEADS patients using the wealth of information documented in electronic clinical narrative notes. NLP is a method of identifying and extracting information from bodies of text, thereby automating the process of reviewing documents. With the growing volume of electronic health record documentation, NLP is increasingly used for identifying information from free text clinical notes for research and practice applications in healthcare. While no previous studies have reported the use of NLP to identify patients who are INEADS, this method has been used to identify a variety of social factors related to health [16].

Healthcare providers spend up to 50% of their time generating and reviewing documentation [17]. Clinical notes bring together subjective and objective findings, observations, diagnoses, and treatment plans, and also serve as a way of communicating detailed information about patients to other clinicians [18]. Therefore, clinical notes may provide additional information, insight, and details related to advance directives, surrogate decision makers, and decisional capacity beyond what is included in structured data or legal documents, which may not exist or be available immediately to the clinical team at the time of care [19]. The aim of this study was to build and validate an NLP algorithm to identify information about patients in the acute care setting that could be used to assess whether they may be INEADS.

## Materials and methods

### Study dataset and population

The Icahn School of Medicine at Mount Sinai Institutional Review Board (IRB) approved this single IRB study and issued a waiver of informed consent. The study was conducted using the publicly available Medical Information Mart for Intensive Care III (MIMIC-III) database, which included de-identified data for 53,423 distinct hospital admissions among 46,520 unique patients who were admitted to an intensive care unit at Beth Israel Deaconess Medical Center, an academically affiliated tertiary/quaternary care hospital with a Level 1 Trauma Center in Boston, MA, between 2001 and 2012 [20]. After excluding patients <18 years, 23,904 distinct hospital admissions among 17,886 patients were included in our analyses.

The database contains demographic information, vital signs, laboratory results, procedures, medications, discharge disposition, and electronic clinical notes. We included electronic clinical notes in the following categories: case management, consultation, discharge summary, nursing, nutrition, physician, rehabilitation, respiratory, social work, and general, which encompassed admission, death, procedure, and progress notes. We excluded notes in categories that were unlikely to contain information related to INEADS status, such as radiology, pharmacy, and echocardiogram. Of the 418,393 clinical notes included, 70% were nursing, 23% were physician, and remaining note types collectively comprised 7% of the total notes.

### NLP algorithm pipeline

Fig 1 depicts the workflow for developing and evaluating the algorithm in six steps.

(1) **Identifying concepts related to INEADS status.** Our interdisciplinary team of medical, nursing, social work, bioethics, public health, and informatics researchers identified concepts related to the three domains of INEADS: decisional capacity, advance directives, and surrogate decision makers. We then categorized these concepts based to create subcategories within each domain (Table 1). The 17 subcategories were annotated based on whether they indicated the reduced or elevated potential for being INEADS. At each stage of development (identifying concepts, classifying concepts into subcategories, and determining potential for

**NLP Algorithm Creation**

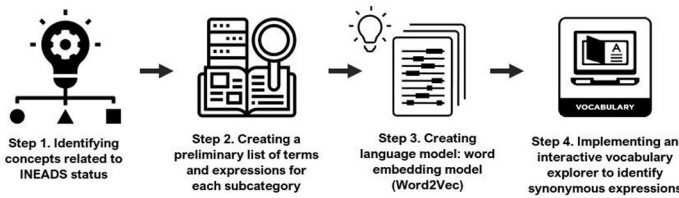

Step 1. Identifying concepts related to INEADS status

Step 2. Creating a preliminary list of terms and expressions for each subcategory

Step 3. Creating a language model: word embedding model (Word2Vec)

Step 4. Implementing an interactive vocabulary explorer to identify synonymous expressions

**Evaluation of NLP algorithm**

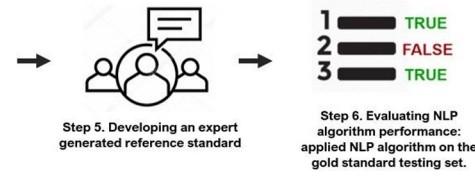

Step 5. Developing an expert generated reference standard

Step 6. Evaluating NLP algorithm performance: applied NLP algorithm on the gold standard testing set.

**Fig 1.**

INEADS), decisions were reached following a comprehensive literature review and group discussion, and culminated in full team consensus.

**(2) Creating a preliminary list of terms and expressions for each subcategory.** To develop the preliminary lexicon for each subcategory, we used a combination of sources including standard clinical terminologies such as the Systemized Nomenclature of Medical Terms (SNOMED) [21], International Classification of Diseases version 10 (ICD-10) [22], and International Classification for Nursing Practice (ICNP) [23]; review of relevant literature; and our team's expertise in nursing, medicine, social work, bioethics, public health, and informatics. Four team members (JS, BC, AL, MT) developed the initial list of terms, which was iteratively revised and finalized by all team members.

**(3) Creating language model: Word embedding model (Word2Vec).** Language models, which are vectorized (numeric) representations of texts in specific domains, allow multiple NLP tasks, such as synonym detection and lexicon creation. We used NimbleMiner [24], an open-source publicly available NLP application in RStudio (Foundation of Statistical Computing, Vienna), to create a language model called Word2Vec algorithm [25] and identify synonyms for the list of preliminary terms in each subcategory within a large body of MIMIC-III clinical notes. This software was selected for its ability to quickly and efficiently identify synonyms from large bodies of text. The system can be downloaded from http://github.com/mtopaz/NimbleMiner under General Public License v3.0.

Briefly, the process of building vocabularies within NimbleMiner is to 1) import a large body of relevant text (clinical notes from MIMIC-III) and a preliminary list of terms for a concept of interests (list of terms within each INEADS subcategory), and 2) perform text preprocessing including deleting punctuations or stop words such as "the" or "is."

**(4) Implementing an interactive vocabulary explorer to identify synonymous expressions.** Using NimbleMiner, we implemented an interactive rapid vocabulary explorer in which the software suggested similar words and expressions based on the language model built in the previous step. For example, when the system was queried for synonyms of the term "palliative care," the system output included terms such as "palliative approach" and "pursue palliative care." Two reviewers (JS and BC) decided whether to accept or reject the words by selecting them in the interactive vocabulary explorer user interface. This process was repeated until no new relevant synonyms were identified. **Table 1** lists examples of terms in each category. The complete list is available in S1 Appendix.

**(5) Developing an expert generated reference standard for evaluation of NLP algorithm performance.** To evaluate NLP algorithm performance, we identified all admissions in which patients had ICD-10 diagnoses that could be associated with being INEADS, including psychiatric disorders, tuberculosis, hepatitis C, and HIV/AIDS. We created a dataset that included all notes from these patients and then randomly sampled a gold standard testing set of 300. Two members of the research team with expertise in nursing, social work, public health, and informatics (BC and AL) manually reviewed each note to identify information that reflected one or more subcategories. Discrepancies between reviewers were resolved through discussion following input from a third reviewer (JS). The observed interrater agreement at the note level for determining whether the note included any documentation indicating elevated potential for being INEADS was good (Cohen's kappa = 0.7).

**(6) Evaluating NLP algorithm performance.** We applied the NLP algorithm on the gold standard testing set to calculate precision (akin to positive predictive value, calculated as [true positives] / [true positives plus false negatives]), recall (akin to sensitivity, calculated as [true positives] / [true positives plus false positives]), and F-score (the weighted harmonic mean of the precision and recall).

**Table 1. Domains and subcategories of terms related to INEADS (Incapacitated with No Evident Advance Directives or Surrogates) status.**

| Domain | Subcategory (sample terms) | Examples of notes containing subcategories | Potential for being INEADS |
|---|---|---|---|
| Surrogate decision makers | Married (husband, wife, spouse) | "…surgical consent signed by *husband*" | Reduced |
| | Partnered (girlfriend, significant other, fiancé) | "Pt [patient] *fiancé* phoned x 1 over noc [nurse on call]…" | Reduced |
| | Living relatives with unknown involvement (son, aunt, complicated family dynamics) | "Family history: *brothers* with [name] in their 50s…" | Reduced |
| | Caregiver support (primary caregiver, sole caregiver, main caregiver) | "social daughter in to visit daughter / lives with pt [patient] and is *primary caregiver*…" | Reduced |
| | Living with or close to others (daughter lives nearby, roommate, lives with) | "Social history: *lives with roommate* works in group home…" | Reduced |
| | Community connection (supportive neighbors, friend who assists, senior center) | "skin grossly intact no breakdown noted. Social mother and *many friends in to visit and supportive*…" | Reduced |
| | Religious connections (their rabbi, clergy involved, practicing catholic) | "current plan of care call *chaplain* in am and set up time for tomorrow evening for *sacrament of the sick*…" | Reduced |
| | Surrogate decision maker identified (family meeting occurred, spoke with daughter, family decided) | "patients critically ill state at this time the *patient family was called…the family decided to* withdraw support…" | Reduced |
| | Unmarried (recently widowed, estranged from his wife, never married) | "social history: works as police officer lives alone *never married* no children…" | **Elevated** |
| | Living alone (lives alone, lives independently, without caregiver) | "…new trauma pt [patient] on tsicu [trauma/surgical intensive care unit] / pt is sp [status post] motorcycle collision with car / pt is a 42 years old man who *lives alone* / in hospital pt did not have demographic information…" | **Elevated** |
| | Transitionally situated (currently homeless, half-way house, incarcerated) | "social ETOH abuse ivda [intravenous drug abuse] *homeless now*…" | **Elevated** |
| | Surrogate decision maker unidentified (cannot find family, family member did not respond, family unreachable) | "social dispo [disposition] full code *no contact from family* per liver notes patient is not a candidate for transplant…" | **Elevated** |
| Advance directives | Palliative care (ethics palliative care, palliative care consulted, palliative services) | "called out to floor plans to have family meeting c [with] team gi [gastroenterology] oncology *palliative care present* emotionally support pt and family" | Reduced |
| | Hospice (moving toward hospice, arrange home hospice, hospice nurses) | "…currently cmo [care management organization] plan to *discharge home with hospice*.." | Reduced |
| | Advance directives available (sister hcp, contact daughter hcp, code status DNR/DNI) | "mother telephone fax 1 *health care proxy appointed yes*…" | Reduced |
| | Advance directives unavailable (no living will, no advance directives, no healthcare proxy) | "…..*no hcp [health care proxy]* lack copy provided…" | **Elevated** |
| Decisional capacity | Lacking capacity (aox disoriented, loss of executive function, impaired judgement) | "she suffered from severe *dementia* along with inability to engage in activities of daily living" | **Elevated** |

## Application of the NLP algorithm and descriptive data analysis

We applied the NLP algorithm to the full dataset of 418,393 clinical notes to identify notes with mentions of the previously described synonyms lexicon. The software also detected negations such as "no" and "denying" to determine the semantic meaning of a sentence. Next, we calculated the proportion of clinical notes that included words or expressions any subcategory and calculated the frequency of each subcategory at the hospital admission level. To establish a

cohort of "possibility of being INEADS" patient admissions, we identified admissions that included at least one of the five social subcategories indicating elevated potential for being INEADS (*unmarried*, *living alone*, *transitionally situated*, *surrogate decision maker unidentified*, and *advance directives unavailable*). We also established a "high likelihood of being INEADS" cohort consisting of admissions that included at least four of these five subcategories. We further bifurcated this group based on whether or not the admission included documentation of at least one subcategory indicating reduced likelihood of being INEADS. We validated the "high likelihood of being INEADS" classification by conducting a case study validation on eight admissions in each segment of this cohort, in which two reviewers (BC and AL) reviewed all notes documented during the admission to determine whether the patient appeared to meet the definition of INEADS. Lastly, we compared demographic characteristics of the "possibility of being INEADS" cohort and the "high likelihood of being INEADS" cohort against the full cohort using t-tests for continuous data or chi-square tests for categorical data. All analyses were performed using R software version 4.1.0 (Foundation of Statistical Computing, Vienna).

## Results

### Cohort demographics

The average patient age was 61 years, 54% were female, and 70% were white. The most common type of insurance was Medicare (59%). During the period captured in the database, 45% of patients died within 4 years after discharge, and 10% died during hospitalization (**Table 2**).

### Validation of NLP algorithm on gold standard testing set

The algorithm demonstrated good performance overall in identifying subcategories compared with gold standard manual review (average F-score = 0.83), with best results for the *transitionally situated*, *palliative care*, and *hospice* subcategories (F-score = 1) and poorest results for *surrogate decision maker unidentified* (F-score = 0.4; **Table 3**).

### NLP detection of subcategories in full cohort

Examples of notes containing each subcategory are provided in **Table 1**. At the note level, 50% of clinical notes (209,697/418,393) contained at least one subcategory. Consistent with the proportion of each type of note in the dataset overall, 68% of notes containing at least one subcategory were nursing notes, 18% were physician notes, and 14% were other note types. However, subcategories indicating elevated potential for being INEADS were most likely to be documented in consult notes (57%) and least likely to be documented in nursing notes (48%).

At the admission level, 81% of admissions (19,380/23,904) had clinical notes that included at least one subcategory. The most frequently documented subcategory was *living relatives with unknown involvement* (72%) followed by *lacking capacity* (57%), *surrogate decision maker identified* (41%), and *advance directives available* (36%). **Fig 2** shows the frequency of admissions with clinical notes that included each subcategory. Seventy-six percent of admissions (18,192/23,904) included at least one subcategory indicating the presence of advance directives or surrogates.

Thirty percent of admissions (7,134/23,904) contained at least one social subcategory indicating elevated potential for being INEADS (*unmarried*, *living alone*, *transitionally situated*, *surrogate decision maker unidentified*, and *advance directives unavailable*) and were included in the "possibility of being INEADS" cohort. Less than one percent of admissions (n = 81) met the criteria for the "high likelihood of being INEADS" cohort, with 79 containing four and two

**Table 2. Characteristics of full, "possibility of being INEADS" and "high likelihood of being INEADS" cohorts.**

| | All admissions | Admissions with possibility of being INEADS (≥1 social subcategory indicating elevated potential for being INEADS) | | Admissions with high likelihood of being INEADS (≥4 social subcategories indicating elevated potential for being INEADS) | |
|---|---|---|---|---|---|
| N (% among all admissions) | 23,904 (100) | 7,134 (29.8) | | 81 (0.3) | |
| Expired during follow-up, n (%) | 10,772 (45) | 3,743 (52) | ** | 53 (65) | ** |
| Expired in hospital, n (%) | 2,479 (10) | 981 (14) | ** | 14 (17) | * |
| Age, mean (SD) | 60 (18) | 61 (19) | * | 61 (15) | * |
| Gender, n (%) | | | ** | | ** |
| Female | 13,000 (54) | 3,793 (53) | | 34 (42) | |
| Male | 10,904 (46) | 3,341 (47) | | 47 (58) | |
| Ethnicity, n (%) | | | ** | | ** |
| Black | 3,304 (14) | 880 (12) | | 11 (14) | |
| Hispanic | 1,014 (4) | 275 (3) | | 2 (2) | |
| White | 16,899 (71) | 5,162 (72) | | 56 (69) | |
| Other | 920 (4) | 245 (3) | | 2 (2) | |
| Unknown | 1,767 (7) | 572 (8) | | 10 (12) | |
| Admission Type, n (%) | | | ** | | ** |
| Elective | 2,788 (12) | 741 (10) | | 5 (6) | |
| Emergency | 20,629 (86) | 6,200 (87) | | 74 (91) | |
| Urgent | 487 (2) | 193 (3) | | 2 (2) | |
| Insurance, n (%) | | | ** | | ** |
| Medicaid | 3,243 (14) | 1,006 (14) | | 16 (20) | |
| Medicare | 14,124 (59) | 4,271 (60) | | 49 (60) | |
| Private | 5,341 (22) | 1,568 (22) | | 12 (15) | |
| Self-pay | 308 (1) | 75 (1) | | 2 (2) | |
| Unspecified government | 888 (4) | 214 (3) | | 2 (2) | |

INEADS, Incapacitated with No Evident Advance Directives or Surrogates.

T-tests (continuous variables) or chi-square tests (categorical variables) compare admissions in the "possibility of being INEADS" and "high likelihood of being INEADS" cohorts with the full cohort.

*p<0.05

**p<0.001

containing all five of the subcategories. Among the "high likelihood of being INEADS" cohort, 10% (n = 8) contained no subcategories indicating reduced likelihood of being INEADS. In our case study validation, all admissions in this group appeared to meet the definition of INEADS following manual note review. Among the remaining 90% (n = 73) of the "high likelihood of being INEADS" cohort, in which admissions contained at least one subcategory indicating reduced likelihood of INEADS, our case study validation of eight cases found that n = 2 (25%) appeared to meet the definition of INEADS.

Table 2 compares admission characteristics across the three cohorts. Admissions in the "possibility of being INEADS" cohort were similar to the full cohort but significantly more likely to die during hospitalization and during the follow-up period. Admissions in the "highest likelihood of INEADS" cohort were significantly more likely to die during hospitalization and during the follow-up period, to have an emergency hospitalization, to have Medicaid (a public insurance for low-income patients), and to be male.

**Table 3. Evaluation of natural language processing algorithm performance through gold-standard manual review (n = 300 clinical notes).**

| Subcategory | Frequency (%) of documentation | Precision | Recall | F-score |
|---|---|---|---|---|
| Living relatives with unknown involvement | 128 (69.6%) | 0.52 | 0.97 | 0.68 |
| Lacking capacity | 119 (64.7%) | 0.67 | 0.99 | 0.8 |
| Surrogate decision maker identified | 53 (28.8%) | 0.94 | 0.75 | 0.83 |
| Advance directives available | 33 (17.9%) | 0.94 | 0.92 | 0.93 |
| Community connection | 20 (10.9%) | 0.85 | 1 | 0.92 |
| Married | 11 (6%) | 0.64 | 1 | 0.78 |
| Unmarried | 7 (3.8%) | 1 | 0.64 | 0.78 |
| Living alone | 7 (3.8%) | 0.86 | 0.75 | 0.8 |
| Living with or close to others | 7 (3.8%) | 1 | 0.64 | 0.78 |
| Surrogate decision maker unidentified | 7 (3.8%) | 0.29 | 0.67 | 0.4 |
| Partnered | 6 (3.3%) | 1 | 0.86 | 0.92 |
| Transitionally situated | 5 (2.7%) | 1 | 1 | 1 |
| Religious connections | 3 (1.6%) | 0.67 | 1 | 0.8 |
| Palliative care | 2 (1.1%) | 1 | 1 | 1 |
| Hospice | 1 (0.5%) | 1 | 1 | 1 |
| Caregiver support | - | - | - | - |
| Advance directives unavailable | - | - | - | - |
| **Overall performance** | 184 (100%) | **0.83** | **0.88** | **0.83** |

## Discussion

In this study, we investigated whether NLP could be applied to identify INEADS characteristics in clinician notes documenting over 23,000 patient admissions to critical care over a 12-year period. To our knowledge, this is the first study to examine the prevalence of INEADS in a large cohort of patients, and the first to do so using electronic health records. Based on the information identified using our NLP algorithms, we found that about half of clinical notes contained at least one subcategory of information that could help determine INEADS status by indicating either reduced or elevated potential for being INEADS.

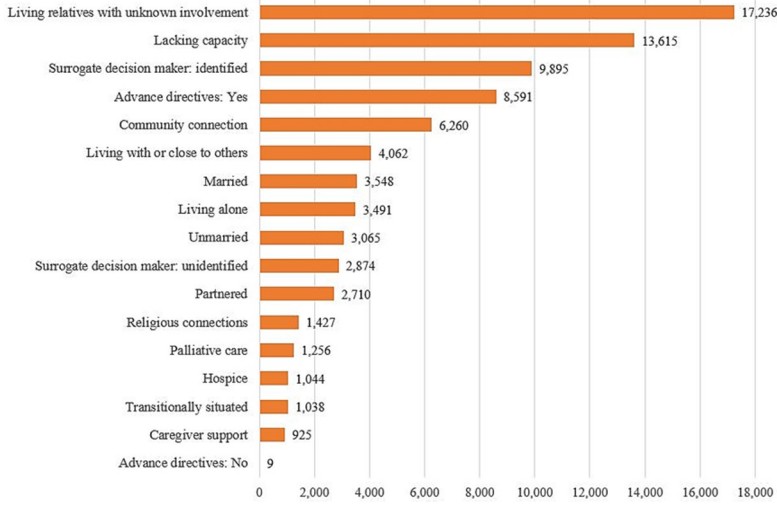

**Fig 2.**

While the algorithms performed well in most categories (F-score = 0.83), they struggled to accurately categorize *living relatives with unknown involvement* and *surrogate decision makers unidentified*, both of which are critically important factors in determining INEADS status. Acknowledging these challenges, our findings indicate that 30% of admissions included at least one characteristic that could indicate risk of being INEADS ("possibility of being INEADS"), while <1% included at least four ("high likelihood of being INEADS"). Patients in both groups were significantly more likely to die during hospitalization and during follow-up, though this may be, in part, an artifact of documentation practices, (i.e., if a patient appears likely to die, the clinical team may document more information regarding advance directives and family or friends who might serve as surrogate decision makers). Male patients were also overrepresented in both groups, which is consistent with previous findings that men are more likely to be socially isolated than women and women have more regular healthcare providers and greater likelihood of having advance directives [26, 27]. Patients in the latter group were significantly more likely to have Medicaid insurance and emergency admissions. This is consistent with previous work showing an inverse relationship between income and advance directive completion, and lower completion rates among patients who are not anticipating critical illness in the near term [2, 28].

Though we found only a small percentage of patients to be INEADS, the absolute number of patients meeting these criteria in the United States is significant [9]. This phenomenon is worthy of further study, especially considering that decision-making processes for INEADS patients are not well characterized, and available data suggest they may be ethically troubling. Two studies by White et al. [14, 15] describe how clinical decisions are made for INEADS patients. In these data, physicians consulted an ethics committee, multidisciplinary review committee, patient advocate, or ombudsman in only 8 of 61 life support withdrawal decisions. Though limited to ICU patients for whom life-support withdrawal was being considered, and collected only from the perspective of the attending physicians caring for them, these studies suggest that physicians may make unilateral treatment decisions for INEADS patients despite recommendations for ethics consultations.

Ideally, when patients lack decisional capacity and advance directives, the care team is able to identify a default surrogate who knows the patient's values, beliefs and preferences, and can offer a substituted judgment of what the patient would want [29, 30]. However, with more adults living in social isolation [31], the healthcare system may face a growing number of patients who will have no voice in determining which life-prolonging or life-limiting care they will receive. Our findings provide important insight into the scope of social considerations among critically ill patients and the ethical challenges that may arise when patients lack decisional capacity, surrogate decision makers, and advance directives. Further studies should work to clarify subcategories and their associations with care needs, and potentially provide early warning signals for intervention, such as social work involvement or ethics consultation, which could have some benefit for patients who have decisional capacity when they enter the hospital and later become incapacitated. The findings of this study also highlight the need for further research to identify patients at highest risk of becoming INEADS and design targeted interventions to discuss goals of care and potential surrogate decision makers in other care settings, such as home care, community-based organizations, or shelters. The goal of all such efforts should be to enhance patients' autonomy and quality of life by respecting their wishes.

This study has several potential limitations. First, although it contains a diversity of patients, the cohort was from a single academic medical center in a northeastern city. Therefore, NLP algorithms might not be generalizable to other sites and settings due to differences in local patient population, documentation practices, and terminology differences. While the algorithm is intended to have applicability across health systems, local validation would be required to determine the algorithm's performance and identify modifications for improvement in each

new setting. Second, the study is limited by its relatively small number of annotated notes used for validation, which may have resulted in our models missing rare but relevant expressions in the entire set of clinical notes. Third, though the subcategories were developed by a team of experts and iteratively refined during the development process, there may be some ambiguity and heterogeneity among the types of expressions identified within each category. For example, the most prevalent category, *living relatives with unknown involvement*, likely includes mention of relatives who are actively involved in care decisions and relatives who are uninvolved, but there was not enough documentation for the NLP algorithm to further specify the level of involvement. In addition, documentation in the *palliative care* and *hospice* subcategories were classified as indicating reduced potential for being INEADS because generally these decisions occur with input and consent from the patient or surrogates. However, in some instances, the NLP algorithm may have detected documentation in which the care team referred a patient to these services due to a need for more in-depth coordination of advanced care planning. Fourth, our method of identifying patients who are INEADS is inherently limited to what the healthcare team documents in clinical notes. Given the diagnostic and physiological emphasis of documentation in the acute care setting, it is possible that factors related to INEADS status were not documented routinely. This could have led to an overestimation or underestimation of the number of patients who may be INEADS. Lastly, our study was limited to information contained within clinical notes and excluded data contained within structured electronic health record fields. Future studies should explore the use of NLP in conjunction with structured or semi-structured data as complementary approaches.

## Conclusion

This investigation demonstrates the potential for NLP to identify patients who may be INEADS, and may inform interventions to enhance advance care planning for patients who lack social support. Efforts to increase completion and portability of advance directives should include strategies for reaching patients who may not have obvious default surrogates, such as those who live alone or in transitional situations. Our novel application of NLP to gather information about social isolation from electronic clinical notes may have implications for healthcare organizations striving to collect and utilize data on social determinants of health.

## Supporting information

**S1 Appendix. Complete list of terms and expressions in each subcategory.**
(XLSX)

## Author Contributions

**Conceptualization:** Jiyoun Song, Maxim Topaz, Robert Klitzman, Jingjing Shang, Patricia Stone, Bevin Cohen.

**Data curation:** Jiyoun Song, Maxim Topaz, Aviv Y. Landau, Margaret McDonald, Bevin Cohen.

**Formal analysis:** Jiyoun Song, Maxim Topaz, Aviv Y. Landau, Jingjing Shang, Margaret McDonald, Bevin Cohen.

**Funding acquisition:** Bevin Cohen.

**Investigation:** Maxim Topaz, Bevin Cohen.

**Methodology:** Jiyoun Song, Maxim Topaz, Jingjing Shang, Patricia Stone, Bevin Cohen.

**Project administration:** Bevin Cohen.

**Software:** Jiyoun Song, Maxim Topaz.

**Supervision:** Patricia Stone, Bevin Cohen.

**Validation:** Robert Klitzman, Margaret McDonald, Bevin Cohen.

**Writing – original draft:** Jiyoun Song, Maxim Topaz, Bevin Cohen.

**Writing – review & editing:** Jiyoun Song, Maxim Topaz, Aviv Y. Landau, Robert Klitzman, Jingjing Shang, Patricia Stone, Margaret McDonald, Bevin Cohen.

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
