## [Decision Letter · Decision Letter 0]

18 Mar 2022

PONE-D-22-04209Using Natural Language Processing to Identify Acute Care Patients Who Lack Advance Directives, Decisional Capacity, and Surrogate Decision MakersPLOS ONE

Dear Dr. Cohen,

Thank you for submitting your manuscript to PLOS ONE. After careful consideration, we feel that it has merit but does not fully meet PLOS ONE’s publication criteria as it currently stands. Therefore, we invite you to submit a revised version of the manuscript that addresses the points raised during the review process.

We look forward to receiving your revised manuscript.

Kind regards,

Giovanni Ottoboni, Psy, PhD

Academic Editor

PLOS ONE

Journal Requirements:

"This study was funded by National Institute of Nursing Research [NINR] (R21 NR019319), “Improving ethical care for patients who are incapacitated with no evident advance directives or surrogates (INEADS)”. The content is solely the responsibility of the authors and does not necessarily represent the official views of the National Institute of Nursing Research. The funders had no role in study design, data collection and analysis, decision to publish, or preparation of the manuscript."

"Our study was funded by National Institute of Nursing Research [NINR] (R21 NR019319), “Improving ethical care for patients who are incapacitated with no evident advance directives or surrogates (INEADS)”. The funders had no role in study design, data collection and analysis, decision to publish, or preparation of the manuscript."

Reviewers' comments:

Reviewer's Responses to Questions

**Comments to the Author**

1. Is the manuscript technically sound, and do the data support the conclusions?

Reviewer #1: Yes

Reviewer #2: Partly

2. Has the statistical analysis been performed appropriately and rigorously? 

Reviewer #1: Yes

Reviewer #2: Yes

3. Have the authors made all data underlying the findings in their manuscript fully available?

Reviewer #1: Yes

Reviewer #2: Yes

4. Is the manuscript presented in an intelligible fashion and written in standard English?

Reviewer #1: Yes

Reviewer #2: Yes

5. Review Comments to the Author

Reviewer #1: This is a very interesting and important study on the use of natural language processing to identify acute care patients who lack advance directives, decisional capacity, and surrogate decision makers. The following points would enhance the study:

Methods:

Study dataset and population: could the authors please provide more information regarding the hospital in which the data is collected (e.g. university or general hospital etc.)?

If patients for instance completed an advance directive with the general practitioner or another primary care provider, might there be a chance that this is included in the clinical hospital notes? Are there shared electronic patient files available between different settings?

Results:

The authors report that 45% of patients died within 4 years after discharge, did the authors collect any information regarding the setting where patients died? Was this data collected in the clinical notes of the hospital? Was this data also collected for those who died outside the hospital? This might need some clarification in the methods section.

Discussion:

The authors found that only a small percentage of patients were characterized as INEADS, could the authors please provide explanations for this low percentage? Might there be an underestimation of this phenomenon?

Could the authors please provide more information on whether this algorithm can be used in other hospitals and healthcare contexts? Should the algorithm be validated again prior using it in for instance another hospital?

Reviewer #2: The manuscript “Using Natural Language Processing to Identify Acute Care Patients Who Lack Advances Directives, Decisional Capacity, and Surrogate Decision Makers” presents an application of NLP on electronic health records to identify patients who are incapacitated with no evident advance directives or surrogates (INEADS).

The authors used an existing open-source dataset including electronic health records and used an NLP algorithm to identify patients who are Incacapacitated with No Evident Advance Directives or Surrogates (INEADS). They developed the NLP algorithm starting from the identification of concepts related to INEADS status (made by three researchers, authors of the manuscript), and, from these concepts, they extracted different subcategories, including different lists of terms. Each subcategory was labeled based on the meaningfulness of that information to classify a patient as an INEADS or not, using labels “reduced” and “elevated”. Then the NimbleMiner algorithm was applied to create a language model, and an interactive vocabulary explorer was implemented to identify synonyms. After a labeling phase from three experts, NLP algorithm performances were evaluated on a (very reduced) gold standard testing set. Once the performances have been assessed, the NLP algorithm has been used on the full cohort to detect the presence of the pre-identified subcategories. Two groups have been detected: admissions with the possibility of being INEADS, having words belonging to more than 1 subcategory, and high likelihood of being INEADS, having words belonging to more than 4 subcategories.

As stated by the authors, this is the first study using NLP to identify INEADS patients. The methodology to use NLP algorithms on electronic health records is well known. The authors of this research used a simple NLP method but other more complex methods could have been investigated and applied. The authors should justify the reason why other methods were not tested and only NimbleMiner was used, or at least present a more comprehensive state-of-the-art regarding NLP and electronic health records.

Secondly, authors stated that “the purpose of the study was to develop and validate a natural language processing (NLP) algorithm to identify patients who are INEADS using documentation in electronic clinical notes”. This is partially true: the applied NLP method is meant to find words belonging to specific categories whose presence is likely to identify INEADS patients. Authors use an arbitrary threshold for the number of the identified categories and, based on the number of identified categories, patients are labeled with a high or low risk of being INEADS. The validation has been made on a very small dataset (10% sample, corresponding to 8 patients), and the authors stated that those patients “appeared to meet the definition of INEADS”, but it is not possible to be completely sure about it, since this information was not reported in the original dataset.

Also, the authors stated that their findings underline the importance of encouraging patients to complete advance directives, particularly when they lack social support. This is not that straightforward from the results, although it can be deduced by comparing the high likelihood of being INEADS demographic with the full cohort. However, this does not seem to be the core of the study, based on what the authors stated (purpose of the study: develop and validate an NLP algorithm to identify patients who are INEADS using documentation …). I would suggest the authors clarify the purpose of the study (develop and validate an NLP algorithm, or investigate the characteristics of INEADS patients?) and be consistent with the conclusions.

Abstract and introduction

Key findings are well reported, but no literature regarding the same topic (i.e., NLP and INEADS patients) was provided. Later in the manuscript, the authors stated that this is the first study attempting to develop NLP algorithms to detect INEADS patients. However, the introduction should include more information about the application of NLP using electronic health records to make the reader better understand what NLP is meant to do. Please add this information in the abstract and the introduction and revise accordingly.

Figures and tables

All the figures and tables are clear and readable. The captions of both figures and tables are complete and accurate. I would suggest adding a different graphical representation for the t-test and chi-square test for clarity.

Methods

I would not use “NLP methods” as the name of the subsection. Instead, I would use “NLP algorithm pipeline”. The word “methods” can be misleading since it seems that more than one method is developed when only one is presented.

Page 10, line 15: “We then created subcategories within each domain”. How were the subcategories created? No information is given. Please, explain in a more extensive way revise accordingly.

Page 10, line 16: “The 17 subcategories were grouped based on whether they indicated reduced or elevated potential for being INEADS”. Instead of “grouped”, I would suggest to the authors to use “annotated” or something similar, since “reduced” or “elevated” is a characteristic of that subgroup. Also, please change “they indicated reduced…” to “they indicated the reduced …”.

Page 11, line 1: “Four team members (JS, BC, AL, MT) developed the initial list of terms, which was iteratively revised and finalized by all team members. Table 1 lists examples of terms in each category”. I would suggest giving the full list of terms within each category as supplementary materials for reproducibility purposes, since only a few examples are available in the main text.

Page 12, line 3: “To evaluate NLP algorithm performance, we randomly sampled a gold standard testing set of 300 clinical notes from all clinical notes of patients with diagnoses that could be associated with being INEADS”. How has the gold standard testing been made? The 300 clinical notes were chosen randomly, or did the authors use a specific rule? Please add information about the note selection in the manuscript.

NLP algorithm performances were evaluated in detecting the 17 subcategories, not in detecting a potential INEADS patient. For this reason, maybe the main research question should be rephrased (INEADS patients are detected based on the arbitrary number (4) of identified categories with an elevated potential of being INEADS).

Page 12, line 23: “…, we identified admissions that included at least one of the five subcategories indicating elevated potential for being INEADS”. The subcategories indicating the elevated potential of being INEADS presented in Table 1 are 6. The “lacking capacity” subcategory is missing in this part of the methods. Only 5 subcategories indicating elevated potential of being INEADS are mentioned throughout the papers. Change it accordingly to what is reported in Table 1.

Page 13, line 2: “surrogate decision maker unknown”. Be consisted of what is written in Table 1, “surrogate decision maker unidentified”.

Page 13, line 4: “We validated the “high likelihood of being INEADS” classification by conducting a case study validation on 10% of this cohort”. Here the 10% means 8 hospital admissions. It is a very small number for the results to be significant. I would suggest validating the classification on a higher number of hospital admissions.

Page 13, line 8: “Lastly, we compared demographic characteristics of the “possibility of being INEADS” cohort and the “high likelihood of being INEADS” cohort against the full cohort using t-tests or chi-square tests”. For the sake of clarity, I suggest to the authors to add specifications on which kind of data (i.e., continuous variables or categorical data) these two statistical methods are applied.

Results

Page 13, line 17: “The most common type of insurance was Medicare”. For a non-American audience, it may be helpful to better explain what “Medicare and Medicaid” stand for (the Medicaid is mentioned in the discussion section).

Page 13, line 21: I would change “NLP algorithm performance against gold standard” in “Validation of NLP algorithm on gold standard testing set”.

Page 14, line 2: “Consistent with the proportion of each type of note in the dataset overall, 68% of notes containing at least one subcategory were nursing notes and 18% were physician notes”. What about the remaining 14%? Which is the source of the remaining notes?

Page 14, line 4: “However, subcategories indicating elevated potential for being INEADS were most likely to be documented in consult notes (57%) and least likely to be documented in nursing notes (48%)”. Is “consult notes” a synonym of “physician notes”? If yes, I suggest remaining consistent using the same terminology throughout the paper.

Page 14, line 17: “Less than one percent of admission (n=81) met the criteria for the “high likelihood of being INEADS” cohort, with 79 containing four and two containing all five of the subcategories”. Are the subcategories indicating elevated potentials of being INEADS six?

Discussion

Page 16, line 21: “However, with more adults living in social isolation”. Please add a reference for this sentence.

Page 17, line 15: “Second, the study is limited by its relatively small number of annotated notes used for training and testing”. Is it right to use “training” for this type of application? The method has pre-defined rules (words within subcategories) and is manually enhanced by researchers by adding some synonyms, so the algorithm does not seem to be “trained” following the strict definition used in machine learning field (https://docs.aws.amazon.com/machine-learning/latest/dg/training-ml-models.html). The algorithm does not seem to “learn” something from the data, rather the authors set rules to identify words.

Conclusion

Page 18, line 10: “Our results underscore the importance of encouraging patients to complete advance directives, particularly when they lack social support”. This conclusion is not so straightforward, based on the implemented methods. Authors should try to justify better their conclusions based on their results.

6. PLOS authors have the option to publish the peer review history of their article (what does this mean?). If published, this will include your full peer review and any attached files.

Reviewer #1: **Yes: **Kim de Nooijer

Reviewer #2: No

---

## [Author Response · Author response to Decision Letter 0]

26 May 2022

Thank you for your careful review and thoughtful comments. We included a table with point-by-point responses to each comment as an attachment.

---

## [Decision Letter · Decision Letter 1]

7 Jun 2022

Using Natural Language Processing to Identify Acute Care Patients Who Lack Advance Directives, Decisional Capacity, and Surrogate Decision Makers

PONE-D-22-04209R1

Dear Dr. Cohen,

We’re pleased to inform you that your manuscript has been judged scientifically suitable for publication and will be formally accepted for publication once it meets all outstanding technical requirements.

Kind regards,

Giovanni Ottoboni, Psy, PhD

Academic Editor

PLOS ONE

Additional Editor Comments (optional):

Reviewers' comments:

Reviewer's Responses to Questions

**Comments to the Author**

1. If the authors have adequately addressed your comments raised in a previous round of review and you feel that this manuscript is now acceptable for publication, you may indicate that here to bypass the “Comments to the Author” section, enter your conflict of interest statement in the “Confidential to Editor” section, and submit your "Accept" recommendation.

Reviewer #1: All comments have been addressed

Reviewer #2: All comments have been addressed

2. Is the manuscript technically sound, and do the data support the conclusions?

Reviewer #1: Yes

Reviewer #2: Yes

3. Has the statistical analysis been performed appropriately and rigorously? 

Reviewer #1: Yes

Reviewer #2: Yes

4. Have the authors made all data underlying the findings in their manuscript fully available?

Reviewer #1: Yes

Reviewer #2: Yes

5. Is the manuscript presented in an intelligible fashion and written in standard English?

Reviewer #1: Yes

Reviewer #2: Yes

6. Review Comments to the Author

Reviewer #1: (No Response)

Reviewer #2: Dear Authors,

thank you for having revised the manuscript. My comments have been addressed and, in my opinion, the quality of the paper has been improved.

7. PLOS authors have the option to publish the peer review history of their article (what does this mean?). If published, this will include your full peer review and any attached files.

Reviewer #1: **Yes: **Kim de Nooijer

Reviewer #2: **Yes: **Serena Moscato

---

## [Editor Report · Acceptance letter]

30 Jun 2022

PONE-D-22-04209R1 

Using natural language processing to identify acute care patients who lack advance directives, decisional capacity, and surrogate decision makers 

Dear Dr. Cohen:

I'm pleased to inform you that your manuscript has been deemed suitable for publication in PLOS ONE. Congratulations! Your manuscript is now with our production department. 

Kind regards, 

on behalf of

Professor Giovanni Ottoboni 

Academic Editor

PLOS ONE